# Centrosome Positioning in Migrating *Dictyostelium* Cells

**DOI:** 10.3390/cells11111776

**Published:** 2022-05-29

**Authors:** Hellen Ishikawa-Ankerhold, Janina Kroll, Dominic van den Heuvel, Jörg Renkawitz, Annette Müller-Taubenberger

**Affiliations:** 1Department of Internal Medicine I, University Hospital, Faculty of Medicine, LMU Munich, 81377 Munich, Germany; hellen.ishikawa-ankerhold@med.uni-muenchen.de (H.I.-A.); dominic.van@med.uni-muenchen.de (D.v.d.H.); 2Walter-Brendel-Centre of Experimental Medicine, University Hospital, Faculty of Medicine, LMU Munich, 81377 Munich, Germany; 3Biomedical Center Munich (BMC), Department of Cardiovascular Physiology and Pathophysiology, Walter-Brendel-Centre of Experimental Medicine, University Hospital, Faculty of Medicine, LMU Munich, 82152 Planegg-Martinsried, Germany; janina.kroll@med.uni-muenchen.de (J.K.); joerg.renkawitz@med.uni-muenchen.de (J.R.); 4Biomedical Center Munich (BMC), Department of Cell Biology (Anatomy III), Faculty of Medicine, LMU Munich, 82152 Planegg-Martinsried, Germany

**Keywords:** amoeboid cell migration, cAMP, chemotaxis, *Dictyostelium discoideum*, folate, microchannels, micropipette assay, microtubules, 3D matrix

## Abstract

Directional cell migration and the establishment of polarity play an important role in development, wound healing, and host cell defense. While actin polymerization provides the driving force at the cell front, the microtubule network assumes a regulatory function, in coordinating front protrusion and rear retraction. By using *Dictyostelium discoideum* cells as a model for amoeboid movement in different 2D and 3D environments, the position of the centrosome relative to the nucleus was analyzed using live-cell microscopy. Our results showed that the centrosome was preferentially located rearward of the nucleus under all conditions tested for directed migration, while the nucleus was oriented toward the expanding front. When cells are hindered from straight movement by obstacles, the centrosome is displaced temporarily from its rearward location to the side of the nucleus, but is reoriented within seconds. This relocalization is supported by the presence of intact microtubules and their contact with the cortex. The data suggest that the centrosome is responsible for coordinating microtubules with respect to the nucleus. In summary, we have analyzed the orientation of the centrosome during different modes of migration in an amoeboid model and present evidence that the basic principles of centrosome positioning and movement are conserved between *Dictyostelium* and human leukocytes.

## 1. Introduction

Cell migration is a complex process and, as such, important for morphogenesis during embryonic development, wound healing, or immune responses. Mechanistically, different types of cell movement can be distinguished: amoeboid, mesenchymal, multicellular streaming, and collective cell migration [1]. Migrating cells are usually characterized by polarity, recognizable by the extension of protrusions at the front or leading edge and retraction of the rear end [2,3].

Individual cells, such as fibroblasts, myoblasts, neural crest cells, and various cancer cells have been shown to migrate in the mesenchymal migration mode, which is characterized by a strong adhesive capacity, mediated by focal adhesions and proteolytic degradation of the surrounding extracellular matrix during interstitial movement. These cells exhibit a rather low migration speed.

In contrast, leukocytes, including neutrophils, T-cells, and dendritic cells, and cells of the model organism *Dictyostelium discoideum* display an amoeboid mode of migration that is characterized by a lack of focal adhesions and stress fibers [4,5,6]. These cells are rather poorly adhesive, migrate with considerably higher speed, and do not perform proteolytic remodeling of their surrounding matrix when migrating in three-dimensional (3D) environments. Depending on the mechanical constraints, amoeboid cells can switch between F-actin-driven and bleb-based pseudopod formation [1,7,8,9,10]. Recent work has shown that both forms of surface extensions can coexist and cooperate during chemotaxis [11,12].

For many years *Dictyostelium* has been used to explore basic principles in cell and developmental biology, and more recently it has emerged as a valuable biomedical model system for studying several human diseases [13,14]. *Dictyostelium* cells are intrinsically motile and serve as an excellent model to analyze cell motility and host defense [13,15,16]. Single *Dictyostelium* cells, just like leukocytes, migrate either randomly or perform directed movement when sensing chemical cues in their microenvironment. During chemotactic migration, the gradient of the extracellular chemoattractant is sensed by the cells through G-protein coupled receptors and transduced into an intracellular signaling cascade, which allows the establishment of cell polarity, expansion of pseudopods, and migration of the cells [17,18,19,20]. Depending on the life cycle phase, either folic acid or cyclic AMP can act as chemoattractant for *Dictyostelium*. *Dictyostelium* cells are professional phagocytes that, in their vegetative growth phase, sense bacteria by chemotaxis toward folic acid and ingest them by phagocytosis [21]. Chemotaxis toward the cAMP released by cells is important for development into fruiting bodies [22]. The cAMP receptor, cAR1, was the first chemoattractant G-protein coupled receptor identified in eukaryotic cells [23], whereas the G-protein-coupled receptor for folic acid-mediated signaling was identified more recently [24].

Microtubules are important cytoskeletal structures, essential for cell division, intracellular transport, motion of cilia and flagella, as well as cell migration and establishment of polarity [25]. How microtubules are involved in the mechanism of migration is not yet fully understood and largely depends on the cell type. Most studies have concluded that microtubules play a positive role, by regulating actin polymerization, transporting membrane vesicles to the leading edge, and/or facilitating the turnover of adhesion plaques. Several studies have shown that microtubules regulate cell migration in a cell type-dependent manner [26,27]. To give a few examples, it was shown that their depolymerization can impair cell migration in types of cells such as fibroblasts [26,28], and suppress the polarity and promote motility of neutrophils [29], while their absence had no influence on the migratory properties of fish keratinocytes [30]. In addition, microtubules have been reported to restrain cell movement and to specify directionality [31]. For immune cells, the role of the microtubule cytoskeleton and its importance in cell polarization and directed migration has been recently reviewed [32].

Microtubules nucleate either from basal bodies or microtubule organizing centers (MTOCs). Centrosomes are the major MTOCs, and during migration, their intracellular position seems to depend on the cell type. In slow-moving non-leukocyte cells, the MTOC is often located in front of the nucleus (relative to the direction of cell migration) and microtubules radiate primarily towards the leading edge [33,34]. Similarly, in slow-moving macrophages, the MTOC often locates in front of the nucleus [35,36]. However, in fast-moving leukocytes, such as dendritic cells and T-cells, MTOCs and microtubules have been described to localize behind the nucleus during directional migration [36,37,38,39]. This positioning of the nucleus, frontward to the centrosome, enables these fast-migrating cells to use their nucleus as a mechanical gauge to probe for suitable larger pores in the microenvironment [39]. However, in fast migrating neutrophils, the MTOC frequently localizes between the lobes of the segmented nucleus [39,40,41], but also has been reported to either localize to the back of the nucleus during polarization on 2D surfaces [39], or to localize to the front of the nucleus during migration in living zebrafish [27], suggesting that MTOC positioning might also be influenced by the cellular microenvironment.

The *Dictyostelium* centrosome has been explored in detail in recent years [42,43]. It is structurally different, as it contains no centrioles and shows some differences with respect to centrosome duplication and its regulation. In the interphase, the *Dictyostelium* centrosome remains adjacent to the nucleus, and several proteins have been demonstrated to play a role in connecting centrosomes to nuclei, including a Sun1 homolog [44,45], a centrin B homolog [46], the centrosomal protein CP148 [47], and the kinesin Kif9 [48]. They play either a structural or regulatory role in anchoring microtubule minus ends into the centrosome corona or in linking components at the nuclear envelope [43,48].

In the present study, we investigated the position of the centrosome relative to that of the nucleus, and the proximity of the centrosome to the nucleus during migration of *Dictyostelium* single cells. Although several studies have addressed the position of the centrosome in the past [49,50,51,52,53,54], a systematic investigation considering different migration conditions has been lacking. Here, we used microfabricated polydimethylsiloxane (PDMS) surfaces and 3D matrices, in addition to standard chemotaxis micropipette assays, to systematically test centrosome positioning in diverse microenvironments and different developmental stages.

## 2. Materials and Methods

### 2.1. Cells and Culture Conditions

Cells of the *Dictyostelium discoideum* strain AX2-214 expressing both GFP-**α**-tubulin (tubA1; DDB0191380|DDB_G0287689) [52] and mRFP-histone (H2Bv3; DDB0231622|DDB_G0286509) [55] were cultivated in polystyrene Petri dishes in HL5 medium (Formedium, Hunstanton, Norfolk, UK) supplemented with 20 µg/mL of Geneticin (Sigma-Aldrich, Sigma-Aldrich Chemie GmbH, Taufkirchen, Germany) and 10 µg/mL of Blasticidin S (Gibco, Fisher Scientific GmbH, Schwerte, Germany) at 22 °C. To induce aggregation competence and development, cells were washed in phosphate buffer (PB; 17 mM phosphate, pH 6.0), and starved in PB for 8 to 10 h.

### 2.2. Chemotaxis Conditions

For chemotaxis of growth phase cells versus pterines, a final concentration of 100 µM folate (Sigma-Aldrich; F8758) was used to set up the gradient in PDMS microfabricated or 3D µ-slide chemotaxis devices (ibidi GmbH, Gräfelfing, Germany) with a VitroGel Hydrogel matrix (TheWell Bioscience Inc., North Brunswick, NJ, USA). For chemotaxis experiments with cells during the aggregation competent stage, cAMP (Sigma-Aldrich; A9501) was used at a final concentration of 10 µM in micropipette assays, PDMS microfabricated devices, and 3D µ-slide chemotaxis devices (ibidi GmbH, Gräfelfing, Germany) with VitroGel Hydrogel (TheWell Bioscience Inc., North Brunswick, NJ, USA) or rat collagen type I (ibidi GmbH, Gräfelfing, Germany) matrices.

### 2.3. Micropipette Chemotaxis Assay

For analysis of cell motility in 2D conditions, starved cells or cells treated with 30 µM of nocodazole for 1 h, were plated in low 35-mm standard-bottom µ-dishes (ibidi GmbH, Gräfelfing, Germany), and migration toward a micropipette (Eppendorf) filled with 10 µM cyclic AMP was recorded using a confocal laser scanning microscope (LSM 780, Zeiss) with a Plan-Apochromat 63x/1.4 Oil DIC objective. Images were taken at 1.5 s intervals for 30–60 min. Centrosome and nucleus displacement were tracked using the automatic ImarisTrack tool of the Imaris software (Bitplane), followed by manual cell tracking.

### 2.4. Migration within Microfabricated Polydimethylsiloxane (PDMS)-Based Microchannels or Pillar Arrays

*Dictyostelium* cells expressing both GFP-tubulin and mRFP-histone were propagated in the growth phase, or the developmental phase as described above (2.1.). Microfabricated PDMS microchannels (8 µm width, 4.8 µm height) or pillar arrays (micropillars with a diameter of 7 µm, positioned with a distance of 10 µm, and a height of 4.2 µm to connect the bottom glass-slide with the PDMS-composed ceiling) were used, as previously described [56,57]. To investigate the chemotaxis of *Dictyostelium* cells in the growth phase, we loaded 100,000 cells and employed a final concentration of 100 µM folate (Sigma-Aldrich, F8758) to set up the gradient. To analyze the chemotactic migration of cells of the developmental stage, 50,000 cells were loaded and 10 µM of cAMP was used to generate the chemoattractant gradient.

### 2.5. Migration in 3D Hydrogel and Collagen Matrices

To analyze the position of the centrosome in relation to the nucleus during migration in 3D environments, 3D µ-slide chemotaxis devices (ibidi GmbH, Gräfelfing, Germany) were used in combination either with VitroGel Hydrogel (TheWell Bioscience Inc., North Brunswick, NJ, USA) or rat collagen type I (ibidi GmbH, Gräfelfing, Germany) matrices.

*Hydrogel:* 100,000 *Dictyostelium* cells of the developmental phase were suspended in PB and mixed with hydrogel (*v*/*v* 1:2), loaded into a 3D µ-slide chemotaxis chamber (6 µL into the middle channel), and were allowed to settle down for 1 h at RT. To set up a gradient of cAMP, a final concentration of 10 µM cAMP was loaded (65 µL) into one outer compartment of the chamber, the other one was filled with PB (65 µL).

*Collagen*: 100,000 *Dictyostelium* cells of the developmental stage were suspended in a final volume of 75 µL containing rat tail collagen type I (ibidi GmbH, Gräfelfing, Germany; 1.5 mg/mL) in PB supplemented with 6.7 mM NaOH, 1.2 mM CaCl_2_, and 0.2% NaHCO_3_. The cell–collagen mix was loaded into the 3D µ-slide chemotaxis chamber (6 µL into the middle channel), and was allowed to settle down for 30 min at RT for polymerization [58]. Then, 100 nM cAMP was added to one side (65 µL) to generate a chemoattractant gradient, and the other side was filled with PB (65 µL).

### 2.6. Live-Cell Imaging of Migrating Cells

Cell migration was recorded using a Zeiss LSM 780 or 880 confocal microscope equipped with a Plan-Apochromat 63x/1.4 Oil DIC objective, image size 512 × 512 pixels, and a frame interval of 1.26 s. For GFP: ex 488 nm/em filter BP 495–550 nm; for mRFP: ex 561 nm/em filter LP 570 nm.

Cell migration in PDMS microchannels and PDMS pillar arrays was recorded with an inverted Leica DMi8 LED fluorescence microscope using an 40× objective, image size 2048 × 2048 pixels, and a frame interval of 10 s. For GFP: ex 475 nm/em 519 nm; for RFP: ex 560 nm/em 594 nm (DFT51011 for both channels).

### 2.7. Statistics

Results of the migration experiments were statistically analyzed using Graph Prism v9 with Students’ *t* tests (Welch’s *t*-test) for Figure 1d, Figure 2d, Figure 3d, Figure 4c,e and Appendix A, or One-way ANOVA (Brown-Forsythe and Welch ANOVA tests) for Figure 1c,f, Figure 2c, Figure 3c, Figure 4b, Appendix A. Data shown represent mean values plus or minus SDs. * *p* < 0.05; ** *p* < 0.01; *** *p* < 0.001.

### 2.8. Determination of Centrosome and Nucleus Distance

Centrosome and nucleus centroids were tracked using an Imaris automatic tool. The track values of each center object (centrosome or nucleus) were subtracted and the difference between the tracks were calculated as the distance of centrosome and nucleus centers (Appendix A). The displayed frames of the trajectory give the values of the nucleus position in relation to the centrosome during oriented migration (Appendix A).

### 2.9. Analysis of the Centrosome Position Nucleus/Centrosome Centroids Relative to Cell Center

*Manual analysis*: Manual analysis of the orientation of the centrosome-to-nucleus axis in linear microchannels (Appendix A) was performed using ImageJ (https://imagej.net/) [59]. Seven horizontal lines with equal spacing were added to each image sequence (Appendix A). Cells that did not cross at least three lines while migrating, were excluded from analysis. Additionally, only single cells migrating directionally along the chemoattractant gradient were analyzed. The nucleus and centrosome length were determined using a line tool from Image J, and the nucleus and centrosome centroids were determined as length/2. The centrosome and nucleus positions, as well as their centroids relative to the cell center, were evaluated for each frame when the nucleus of a cell reached one of the horizontal crossing lines (Appendix A). To calculate the nucleus and centroid distances to the cell center, seven crossing lines were drawn in each image and used as reference for the calculation of cell length, the cell centroid, and the distances from the nucleus/centrosome centroid to the cell center (Appendix A). The nucleus/centrosome centroid distances to the cell rear, where divided by the distance of the cell center to the back of the cell; thus, the values closer to 1.0, mean closer to the center of the cell. The evaluation of centroid centrosome/nucleus distance to the cell center of a single cell per frame was calculated for the folate experiments, as displayed in Appendix A.

For cAMP or folate chemotaxis experiments in linear microchannels (Appendix A), the positions of the nucleus and centrosome along the cell axis were manually determined by measuring their distance from the cell rear in relation to the cell length, to calculate their relative intracellular position along the cell axis (Appendix A). From this dataset, we further calculated the centrosome-to-nucleus distance (Appendix A) and the nucleus/centrosome centroid relative distance to the cell center (Appendix A).

*Automatic analysis*: Centrosome and nucleus centroids were tracked using the Imaris automatic tool “Spots creation tool’s automatic generation feature”. After completion of automatic generation, further optimization of tracks was completed manually. The trajectory was determined between a reference point, where the chemoatractant was released, and the centroid of the objects (nucleus and centrosome). The distance from the centroid centrosome minus the distance of the nucleus centroid to a reference chemoatractant point was calculated and used to determine the centrosome position (Appendix A). A spot of 2.0 µm was set for the nucleus and a spot of 1.0 µm for the centrosome, with centroids determined automatically. The distance of the nucleus centroid to the nucleus border was set to 1.0 µm, and the centrosome centroid distance to the centroid border was 0.5 µm. Thus, all centrosomes with a difference of centroid centrosome distance minus the centroid nucleus larger than +1.5 µm were characterized as having a centrosome position at the ‘back’ of the nucleus. Values smaller than +1.5 µm were characterized as “side-back” position. Positions of the centrosome within differences smaller than −1.5 µm were characterized as “side-front”, and larger than −1.5 µm as “front” (Appendix A).

## 3. Results

### 3.1. Nucleus and Centrosome Positioning in Dictyostelium Cells Migrating in Confined Environments

To study centrosome and nucleus positioning in migrating *Dictyostelium discoideum*, we employed cells in the early developmental stage, stably encoding both the nuclear marker mRFP-histone and GFP-tubulin as a marker for the microtubules and centrosomes, and performed live-cell imaging in linear microchannels with a gradient of cAMP as chemoattractant. In this confined environment, *Dictyostelium* cells migrate highly directionally and persistently along straight paths (Figure 1a; Appendix A). This setup allows the precise quantification of the orientation of the centrosome–nucleus axis with an accurate centrosome positioning classification (Figure 1b and Appendix A), as well as the determination of the nucleus and centrosome centroids relative to the cell center (Appendix A). Automated tracking (Appendix A) and quantification, which we controlled by manual analysis (Appendix A), revealed a preferential positioning of the centrosome behind the nucleus (Figure 1c). While we also observed positioning of the centrosome to the sideward and front of the nucleus (Figure 1c), and dynamic re-positioning of the centrosome closer to the cell center in individual cells (Appendix A), we found a strong preference of the centrosome to be positioned in very close proximity, behind the nucleus (Figure 1c,d).

We next imaged mRFP-histone and GFP-tubulin expressing *Dictyostelium* cells migrating along a cAMP gradient in between arrays of micropillars, as this environment requires *Dictyostelium* to deviate from entirely straight paths, by performing turns around individual micropillars. Notably, we observed that during reorientation of the centrosome-nucleus, the position of the nucleus within the cells stays relatively unchanged, while the centrosome dynamically repositions between a frontward and rearward localization in relation to the nucleus (Figure 1e; Appendix A). Again, we noted a close proximity of the centrosome and the nucleus, and a strong preference of the centrosome to be positioned at the back and side-back of the nucleus (Figure 1f).

To corroborate these results, we investigated the orientation of the centrosome–nucleus axis while *Dictyostelium* migrates within a gradient of cAMP released from a micropipette (Figure 2a,b). The repositioning of the micropipette filled with cAMP, causes a local change within the gradient, which is sensed by the cells and causes reorientation towards the pipette tip within seconds [60]. While *Dictyostelium* cells migrated in this experiment on a two-dimensional surface, their path was not dictated by the microenvironment and, thus, allowed non-straight migration, such as between micropillars (Appendix A). Moreover, in this experimental setting, we observed a non-random orientation of the centrosome–nucleus axis, with the centrosome located behind the nucleus (Figure 2c).

### 3.2. The Role of Microtubules for Centrosome Positioning in Dictyostelium Cells Migrating in 2D Confined Environments

To investigate the role of the microtubule cytoskeleton in centrosome positioning, we tested the orientation of the centrosome–nucleus axis after application of the microtubule-depolymerizing drug nocodazole. Nocodazole interferes with the polymerization of microtubules, and though it is known to be less efficient in *Dictyostelium* than in other eukaryotes, it causes massive shortening of microtubules, visible as short stumps radiating from the centrosome [61]. Live-cell imaging of migration to a cAMP-containing micropipette still revealed a preferential positioning of the centrosome behind the nucleus in the presence of nocodazole, yet with a more frequently inverted orientation, in which the centrosome is positioned frontward of the nucleus (Figure 2c). In the presence of nocodazole, *Dictyostelium* cells migrate with reduced velocity (Appendix A) and show a higher mobility of the centrosome when the microtubule cytoskeleton is non-functional, leading to a reduction of the distance between nucleus-centrosome centroids (Figure 2d). Nocodazole treatment abolishes the contact of long microtubules that typically span through the cell body toward cortical areas, whereas short microtubules in the direct vicinity of the centrosome remain present (Appendix A). Thus, this suggests that the remaining short microtubules mechanically connect the centrosome with the nucleus.

These findings show that a functional microtubule cytoskeleton contributes to the correct orientation of the centrosome–nucleus axis in motile *Dictyostelium*.

### 3.3. Nucleus and Centrosome Positioning in Dictyostelium Cells Migrating in 3D Confined Environments

We then tested whether the observed configuration of the centrosome-to-nucleus axis can be characterized as well in three-dimensional matrices, which represent a close proxy of natural environments. To this end, we imaged chemotactic migration of *Dictyostelium* in hydrogel, as well as collagen matrices (Figure 3; Appendix A). As in 2D migration, *Dictyostelium* cells moving in 3D showed a strong preference to position the centrosome behind the nucleus (Figure 3a; Appendix A). This location was altered upon depolymerization of microtubules (Figure 3b; Appendix A), which resulted in the centrosome being found much more frequently on the side or front-ward of the nucleus (Figure 3c). It should also be noted that, in nocodazole-treated cells, the distance between the centrosome and the nucleus was slightly reduced (Figure 3d).

In summary, we found that the centrosome and the nucleus are non-randomly positioned in motile *Dictyostelium* cells. This non-random configuration positions the nucleus in front of the centrosome, with respect to the direction of migration. This orientation is clearly preferred in motile *Dictyostelium* cells at the developmental stage, occurring on flat two-dimensional substrates, in confining microchannels, in between micropillars, as well as in three-dimensional matrices.

### 3.4. Nucleus and Centrosome Positioning in Dictyostelium Cells Migrating in Confined Environments toward Folate

To test whether the orientation of the “centrosome-nucleus in front” axis is the generally preferred position at the different developmental stages of *Dictyostelium*, we imaged cells at the growth phase migrating chemotactically along a folate gradient in linear microchannels (Figure 4a; Appendix A). Subsequent analysis revealed a strong preference to position the centrosome behind or side-back of the nucleus (Figure 4b). The determination of the distance between the centroids of the nucleus and the centrosome revealed no fundamental differences between cells migrating in folate or cAMP gradients (Figure 4c). The plot of an exemplary track shows how the centrosome follows the nucleus (Figure 4d). In 2D microchannels, growth phase *Dictyostelium* cells migrate more slowly toward folate than cells of the early aggregation state toward cAMP as chemoattractant (Figure 4e). A similar difference has been shown previously for other conditions.

Altogether, our data show that *Dictyostelium* cells position their centrosome and the nucleus in a non-random orientation during motility at different developmental stages, with the preferred position of the centrosome always at the back of the nucleus. This positioning is reinforced by microtubules that emanate from the centrosome and extend to the front cortex. Changes in the direction of a cell caused by a change in the chemoattractant gradient results in a transient displacement of the centrosome from the pseudopod-nucleus-centrosome axis.

## 4. Discussion

The positioning of organelles inside cells is non-random, and this non-random positioning is functionally important [62]. The nucleus — the largest and stiffest cell organelle [63] — and the centrosome — the major microtubule organizing center [25] — are typically positioned in close proximity in a particular orientation [33]. Many motile cells, such as fibroblasts and neurons, preferentially position their centrosome frontward of the nucleus [34], e.g., during polarized fibroblast migration into cell-free wound areas [64]. This particular alignment of the centrosome-to-nucleus axis is established by linking the nucleus with the cytoskeleton [65], and supports local release of proteases for extracellular matrix proteolysis at constricting pores [66], and the pulling of the nucleus through narrow pores [67].

Whereas the cellular microenvironment (e.g., degree of confinement) also appears to influence the orientation of the centrosome-to-nucleus axis [33,68], the general concept emerges that slowly migrating mesenchymal-like cells preferentially position their centrosome in front of the nucleus. In contrast, recent findings have shown that fast amoeboid-like migrating immune cells, such as dendritic cells, preferentially position their centrosome to the rear of the nucleus [38,39]. This unexpected positioning of the nucleus frontward to the centrosome [32,39], and closely behind the cellular leading edge [39,69], supports these fast migrating cells in using their nucleus as a mechanical gauge to probe for suitable larger pores in the microenvironment along their paths of migration [39].

Here, we investigated the positioning of the centrosome-to-nucleus axis in motile *Dictyostelium discoideum* cells. *Dictyostelium* represents a long-standing cellular model to discover general concepts in cell motility [19,20,70,71,72,73]. Surprisingly, the positioning of the centrosome in relation to the nucleus in this traditional model for amoeboid migration was initially analyzed only by evaluating electron micrographs [50]. The authors came to the conclusion that the centrosome shows no preferential position anterior or posterior to the nucleus; rather, its position correlates with the type of migration and perhaps with the nature of cell–cell adhesion [50].

Another early study previously showed that the centrosome does not bias the cell, as it was observed that after formation of a new pseudopod, the centrosome reorientates within an average of 12 s [51]. Thus, they concluded that the centrosome does not determine the alignment of the movement [51]. However, it has been consistently discussed that the centrosome may also be located anterior to the nucleus relative to the direction of movement, depending on the developmental stage or the conditions of migration. In summary, a systematic analysis that investigated the position of the centrosome during migration was lacking.

To close this knowledge gap, we investigated two major motile states of *Dictyostelium:* single-cell migration in the growth phase along a folate gradient, and single-cell migration in the developmental stage along a cAMP gradient. We found that the centrosome is positioned in both states close to the cell center, and that the nucleus is positioned frontward of the centrosome. Furthermore, by extensively investigating *Dictyostelium* migration in diverse microenvironments, ranging from two-dimensional substrates and confining microchannels to three-dimensional extracellular matrices, we discovered that this specific orientation of the centrosome-to-nucleus axis is a general feature of migrating *Dictyostelium* cells. Thus, amoeboid migrating *Dictyostelium* cells position the nucleus frontward of the centrosome, similarly to amoeboid migrating immune cells, suggesting evolutionary conservation of centrosome-to-nucleus positioning in motile amoeboid cells.

The extension of a pseudopod in a new direction causes the reorientation of the centrosome. A crucial question that remains to be solved is how these processes are coupled. The present study was not designed to answer this general question. However, our findings are in agreement with the assumption that reorientation of the centrosome and microtubules extending toward the front reinforce the direction of movement. Thus, the active balancing of the pseudopod-nucleus-centrosome axis is a critical component. Our results with nocodazole-treated cells strongly support this notion.

Our findings suggest a general concept, in which fast moving amoeboid-like cells position their nucleus in front of the centrosome, whereas slow mesenchymal-like cells position their nucleus behind the centrosome. In future, it will be interesting to further elucidate what other functional significance this configuration and positioning of the centrosome–nucleus axis has for amoeboid migration. Moreover, our findings highlight the relevance of the genetically accessible amoeba *Dictyostelium* as a cellular model to discover mechanisms of cell migration. This approach will also help to transfer well established techniques in *Dictyostelium* to the leukocyte model system, and thus may yield new insights into principles of directional cell migration and polarity, as important elements in host defense, wound healing, and development.

## Figures and Tables

**Figure 1 cells-11-01776-f001:**
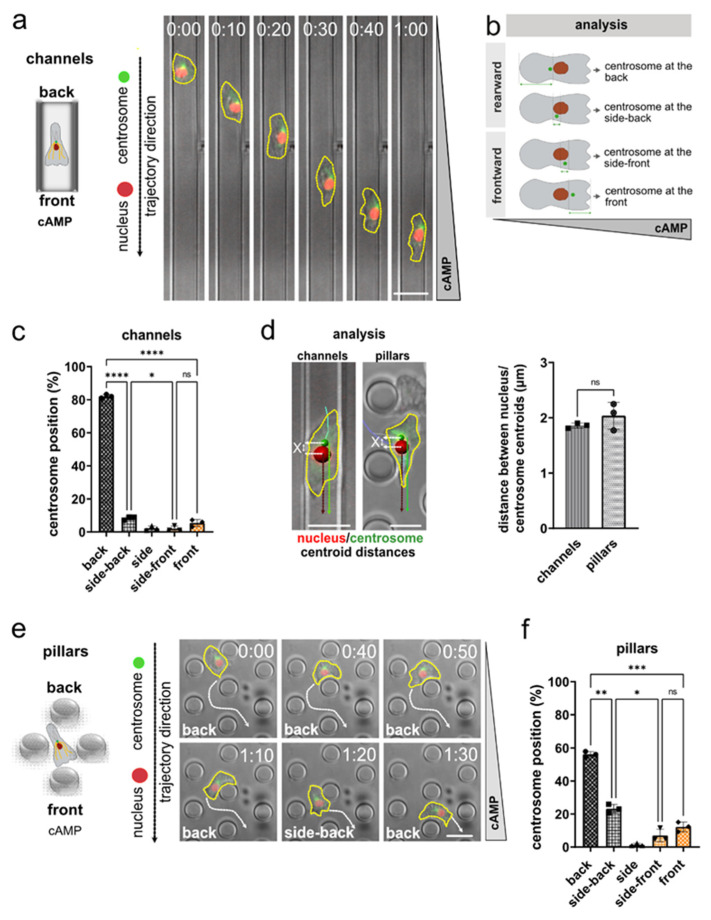
The centrosome of *Dictyostelium* cells migrating in microchannels or micropillar arrays along a cAMP gradient is preferentially located behind the nucleus. (**a**) Scheme (left) and representative microscopy images (right) of an aggregation competent *Dictyostelium* cell migrating in a microchannel along a gradient of cAMP. The cells express both the nuclear marker mRFP-histone (red), and GFP-tubulin (green) to highlight centrosomes and microtubules. Time is indicated in min and s at the top. The cell shape is highlighted by a yellow dashed line. (**b**) Principle of quantification. (**c**) Centrosome position during migration in microchannels within a gradient of cAMP. *N* = 3, number of cells = 32. (**d**) Representative images of *Dictyostelium* cells showing analysis of nucleus (red) and centrosome (green) centroid distances (indicated by “X”) (left). Histogram (right) displays the quantification of the distances between the nucleus and centrosome centroids during migration in microchannels and micropillar arrays along a cAMP gradient. *N* = 3, number of cells = 30. (**e**) Scheme (left), and representative microscopy images (right) of a cell migrating in a field of micropillars along a gradient of cAMP gradient, recorded with a time interval of 10 s per frame. The white arrows indicate the trajectory of the cell. The numbers indicate the time in min and s. (**f**) Quantification of the centrosome position during migration in an micropillar array along a gradient of cAMP. *N* = 3, number of cells = 30. Scale bars are 10 µm. * *p* < 0.05, ** *p* < 0.01, *** *p* < 0.001, **** *p* < 0.0001 are significant, and ns = not significant.

**Figure 2 cells-11-01776-f002:**
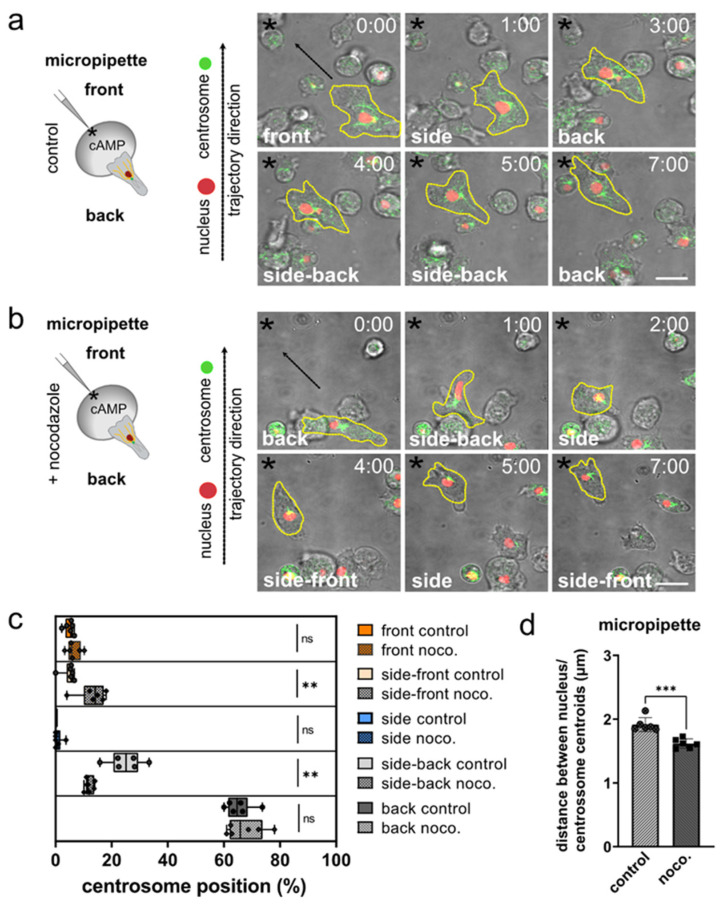
In aggregation-competent *Dictyostelium* cells moving in a gradient of cAMP released from a micropipette, the centrosome is positioned behind the nucleus, but frequently relocates when microtubules are disrupted. (**a**,**b**) Schemes (left) and representative microscopy images (right) of *Dictyostelium* cells migrating toward a micropipette tip releasing cAMP (indicated by asterisks). The cells express both the nuclear marker mRFP-histone (red), and GFP-tubulin (green) to highlight centrosomes and microtubules. Time is indicated in min and s at the top. The cell shape is highlighted by a yellow dashed line. Images were taken with 10 s-frame intervals. *N* = 6, number of cells = 50 for control; *N* = 6, number of cells = 52 for nocodazole. (**a**) Shows control cells, (**b**) nocodazole-treated cells. (**c**) Analysis of centrosome positions in control and nocodazole-treated (noco.) cells. Note that there were no events for the side control. (**d**) Nocodazole treatment decreases the distance between the center position (centroid) of the nucleus and the center of the centrosome. Scale bars are 10 µm. * *p* < 0.05, ** *p* < 0.01, *** *p* < 0.001 are significant, and ns = not significant.

**Figure 3 cells-11-01776-f003:**
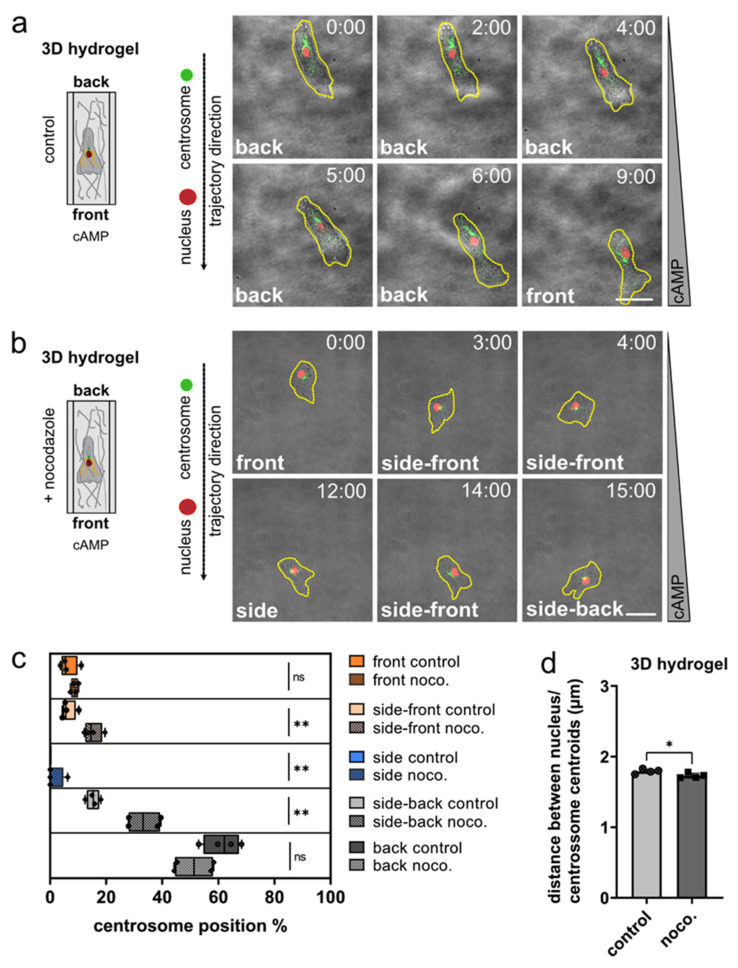
Centrosome positioning in aggregation competent *Dictyostelium* cells moving chemotactically in 3D environments. (**a**,**b**) Schemes (left), and representative microscopy images (right) of a *Dictyostelium* cell migrating in a chemotaxis chamber with a 3D hydrogel matrix. The cells express both the nuclear marker mRFP-histone (red), and GFP-tubulin (green) to highlight centrosomes and microtubules. The cell shape is marked by a yellow dashed line. Images were taken with 10 s-frame intervals. Time is indicated in min and s at the top. *N* = 4, number of cells = 40 for control; *N* = 4, number of cells = 40 for nocodazole. (**c**) Analysis of the position of the centrosome in *Dictyostelium* cells moving chemotactically in 3D hydrogels. Note that there were no events for side control. (**d**) After nocodazole treatment, in cells moving chemotactically in 3D the distance between the centroid of the nucleus and the centroid of the centrosome was slightly decreased. Scale bars are 10 µm. * *p* < 0.05, ** *p* < 0.01 are considered significant, and ns = not significant.

**Figure 4 cells-11-01776-f004:**
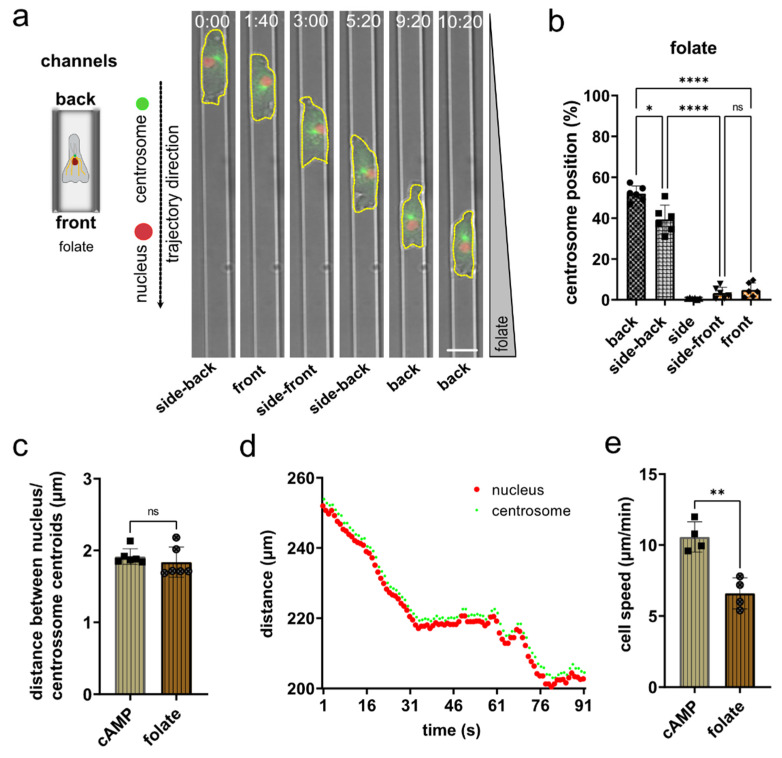
In growth phase *Dictyostelium* cells migrating in microchannels along a gradient of folate, the centrosome is preferentially located behind the nucleus. (**a**) Scheme (left) and representative microscopy images (right) of a *Dictyostelium* cell migrating in a microchannel along a folate gradient. The cell shape is highlighted by a yellow dashed line. Images were taken with 10 s-frame intervals. Time is indicated in min and s at the top. (**b**) Quantification of the centrosome position during migration in microchannels along a gradient of folate. *N* = 6, number of cells = 40. (**c**) Quantification of the distance between nucleus and centrosome of *Dictyostelium* cells migrating in microchannels in a folate gradient (*n* = 6, cells = 40). (**d**) Displacement of tracks for nucleus (red) and centrosome (green) recorded for one cell over time. (**e**) Migration speed in microchannels determined for growth phase *Dictyostelium* cells moving along a gradient of folate and for aggregation competent cells moving in a gradient of cAMP. * *p* < 0.05, ** *p* < 0.01, **** *p* < 0.0001 are significant, and ns = not significant.

## Data Availability

If not contained within the article or Appendix A of this publication, data are available from the corresponding author.

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
