# Peer review of "Centrosome Positioning in Migrating Dictyostelium Cells"

_cells, 2022, doi:10.3390/cells11111776_

Round 1
Reviewer 1 Report
Critical roles that nucleus positioning plays during lymphoid cell migration has been highlighted in recent years. In these cells, nucleus take large fraction of the entire cell volume, and thus positioning and deformation of the nucleus plays substantial part of the cell mechanics especially when the cells need to squeeze through narrow and confined intercellular space. In contrast, Dictyostelium has relatively small nucleus, and how its position and shape are coordinated during cell migration has not been systematically quantified.
The paper entitled “Centrosome positioning in migrating Dictyostelium cells” by Ishikawa-Ankerhold et al analyze the centrome and nucleus positioning in migrating Dictyostelium cells by means of live cell imaging analysis that makes use of D. discoideum cells co-expressing mRFP-histone and GFP-tublin and microchambers to confine cell movement. Overall, the experiments and the analysis are carefully executed and the results are described well. The conclusion is that there is a greater tendency for the centrosome to position itself at the back of the nucleus in migration Dictyostelium cells and that this is observed in a variety of different conditions which is new and important. There are some points I hope authors can address for an additional insight.
Major point:
1) It is interesting that the extension of a pseudopod in a new direction correlates with the centrosome repositioning. I am curious to know whether the nucleus position is biased towards the front/pseudopod of migrating cells. If so, is this altered in nocodazole-treated cells? For example in Fig. 4a (1:40), when the orientation flips, it looks as though the nucleus is at the tail of the cell while the centrosome just remained at the cell center. In general the centrosome has a propensity to position itself at the center of the cell as a result of the force-balance between the polymerizing microtubules. While this may not be the case in Dictyostelium as its microtubles seem to have much shorter persistent length, actin in some systems can also contribute to the nucleus position. Authors can for example measure the positioning of the centrosome and the nucleus from the cell centroid or from the leading edge in some of the examples shown in Fig 3 and Fig. 4 and discuss how this do or do not relate to their relative positioning.
2) I had hard time figuring out how to read Fig. 3c. Is the side control data there or is it missing? I guess what I don’t understand is how to read the x-axis. These are plots of data from differnet conditions so what 100% is not trivial. Please clarify in the figure legend.
Minor points.
I assume this is beta or alpha-tubulin fused to GFP not gamma-tubulin but this was not explicitly stated. If so I assume the authors take the most intense fluorescence area as the centrosome position which is also not described. Also which histone is this?
L.133: 100.000 cells -> 100,000 cells ( I know)
- 135: 50.000 cells -> 50,000 cells
- 169: ‘values for of the’
- 413: period missing after ‘general question’
Fig. S1a I wouldn’t categorize folate as ‘chemokine’. Some of the panels and fonts are too small.
Reviewer 2 Report
The article studies an important problem - the spatial position of the nucleus and centrosome in motile cells. The results of the work are documented in detail and provide adequate substantiated answers to the questions posed. My remarks concern the review of previously obtained data on the problem under study.
Line 85-87
In the introduction and discussion, it is necessary to cite comprehensive work on the study of the position of the centrosome in various cell types in the laboratory of Professor Chentsov of Moscow University.
The centrоsomes in blood cells have a different location with respect to the nucleus and the leading edge of the cell: in macrophages the centrioles are situated mostly anterior to or at the side of the nucleus; in granulocytes they lie between the nuclear segments; and in lymphocytes they are positioned strictly posterior to the nucleus, in the uropod. In each case, however, the centrioles are localized in the central region of the cytoplasm. Their alignment does not appear to be related to the blood cells' random motion in vitro.
Gudima et al. (1984) Cell center of macrophages, granulocytes and lymphocytes during in vitro cell spreading, polarization and movement. Tsitologiia. 1984 Sep;26(9):1002-7. PMID: 6506219
https://pubmed.ncbi.nlm.nih.gov/6506219/
Gudima et al. (1988)
G O Gudima, I A Vorobjev, Chentsov YuS (1988) Centriolar location during blood cell spreading and motion in vitro: an ultrastructural analysis
J Cell Sci . 1988 Feb;89 ( Pt 2):225-41. doi: 10.1242/jcs.89.2.225.
https://pubmed.ncbi.nlm.nih.gov/3263378/
Line 87-89
In neutrophils, according to electron microscopy, the centrosome is located between the lobes of the nucleus (Uzbekov et al., 1989)
R E Uzbekov, I A Vorob'ev, V A Drachev The effect of the laser microirradiation of the cell center on neutrophil motility. Tsitologiia . 1989 Aug;31(8):874-81.
Minor technical notes:
1) You can enter the abbreviation MT for microtubules
2) Minimize word wrapping from line to line.
After the discussed results are supplemented by the indicated publications, the article can be accepted for publication in the journal Cells.
Reviewer 3 Report
The authors H. Ishikawa-Ankerhold et al explored the positioning of centrosome relative to the nucleus in Dictyostelium discoideum cells by live-cell microscopy. Here, the authors analyzed the orientation of the centrosome during different modes of 2D and 3D migration in an amoeboid model. They present numerous evidence that single cells Dictyostelium position their centrosome and the nucleus in a non-random orientation during motility in different developmental stages, and the preferred location of the centrosome is always at the back of the nucleus. As reviewer, I agree to say that their manuscript is quite interesting, in particular because all of their results (both experiments and simulations) could support for the first time directional changes of centrosome caused by a change in the chemoattractant cAMP or folate gradient. However, despite this as well as the quality of the English and writing in the manuscript, many points concerning both the clarity of the manuscript and important details (experimental and scientific) detract from the quality of this study. Thus, all of these preclude acceptance the article for publication in Cells, as presented below.
Major comments:
- in the section Introduction: this section is not correctly written, the problem of the study is not clearly exposed, and too many generalities are given which do not allow the interest of the results to be understood. The Dictyostelium cell model is not a recent and current study model, even though it is very adapted for describing and understanding certain mechanisms involved during migration (in particular, the amoeboid migration); it is therefore crucial that the authors rewrite several passages of this section to better understand the stakes of their interesting work:
1/ lines 53-68: please, specify in what and how G protein-coupled receptors control cytoskeletal remodeling, and ultimately regulate cell polarization and migration
2/ lines 69-82: I agree with the authors in granting MTs an important role in the mechanisms described here, but we cannot ignore the role of F-actin too, it is then common to speak of crosstalk between the two cytoskeletons; so what is the use of making all these reminders about MTs for the study here?
3/ lines 71-72: please, modify the sentence (group of words "is still controversial) since the role of MTs in migration depends on the cell type, therefore it is not "controversial"
4/ line 92: it is a pity that the authors do not wish to recall more about the MTOC and centrosome system in Dictyostelium, whereas this is precisely the model of their study. These reminders would allow the novice to better understand the problem (which is not yet clearly stated at this level of the introduction)
- in the section Materials and Methods:
5/ line 103: which isotype of tubulin was chosen for fluorescent labeling (GFP) of centrosomes? Also, which histone was tagged with mRFP? Please specify this information in the text
6/ lines 154-159: please, complete the description of each microscopy system by specifying the excitation wavelengths and emission filters used, otherwise the chosen emission wavelengths
- in the section Results:
7/ line 221: specify how "the position of the nucleus stays relatively unchanged" in the text, is it in relation to its position between the front and the back of the cell?
8/ Fig 2a-b: please, show the fluorescence and DIC images separately, so as to better observe the position of the centrosome (in green) in relation to the nucleus (in red); moreover, there seems to be many immobile cells (dormant cells?), this affects the clarity of the images: could the authors present a better field of observation?
9/ Fig 2c-d: also, although treatment with nocodazole leads to disturbances in the positioning of the centrosome (only in the side-front and side-back positions), it is difficult to know whether this is really due to the disassembly of the MTs (as clearly showed in the fig S2b revealing shorter MTs after treatment); the authors could answer this question by carrying out a nocodazole wash-out: after reassembly of the MTs, they should observe a recovery of the positioning of the centrosomes comparable to the control condition
10/ lines 289-292: the proteins involved in the attachment of the centrosome to the nucleus are well known (also known as a nucleus associated-body in Dictyostelium), for which mention may be made in particular of dynein or Kif9; moreover, the conclusions are quite poor: the literature showing the role of functional MTs on the orientation of the centrosome-nucleus axis is vast, with models more complex than this one
11/ lines 299-305 + Fig 3c-d: it is true that during 3D migration, the centrosome may position itself to the side of the nucleus more frequently, following nocodazole treatment. However, for 3D migration, we can also observe an increase in the frequency of side-back positions in the presence of nocodazole, contrary to 2D migration (Fig 2c). This is a point that the authors do not discuss, why? Finally, it is possible that detailing so much the "side" position is not relevant, perhaps it would be better to determine three groups of positions: in the back, in the front and on the side of nucleus? The message would probably be simpler.
12/ lines 343-346: please, review the impact of folate in the growth phase migrating of Dictyostelium in order to better understand the objectives of this part
13/ Fig 4: are the authors led to the same observations with folate during the 3D migration? Did they test in the presence of nocodazole? As the cell speed is reduced by 2 times with folate (comparable values with nocodazole) compared to cAMP (in Fig 4e) without modification of the nucleus-centrosome distance (in Fig 4c-d), it is therefore possible that this is the crosstalk between functional MTs and actin-F in the pseudopod which controls the positioning of the centrosome in the nucleus-centrosome axis; it would therefore be interesting to label actin simultaneously with MTs and centrosome under the different experimental conditions tested (for example with immunofluorescence); as well as testing for F-actin disrupting agents; it would also be important to identify the role of dynein in the connection and maintenance of the centrosome to the nucleus. Indeed, dynein provides a related function to connect nucleus and centrosome via force-generating machinery acting on MTs in many metazoan systems: here, the authors could confirm their conclusions (lines 355-361) by using dynein mutants in their experimental conditions
- in the section Discussion: overall, the authors correctly place their results in the context of knowledge on the subject
Minor comments:
- line 95: please, add more recent references to 36-38
- lines 107-108: please, indicate final concentration of phosphate PO4 in buffer
- line 118: does "(2.1.)" refer to section "2.1. Cells and culture conditions" above? What is the use of this precision in the text?
- line 134: please, replace “chemotatic” by “chemotactic”
- line 142: please, specify “(V/V 1:2)”, and not only “(1:2)”
- line 145: please, remove the extra word "with" from the sentence
- line 155: please, specify the resolution of the images in pixels instead of their dimensions
- line 169: please, remove the word “for” in the sentence
- line 205: please, homogenize the “mRFP” nomenclature which is specified by RFP-histone in the M&M section
- Fig 3d: please, remove plots on bars, this affects the readability of the differences between the conditions
Round 2
Reviewer 1 Report
Thank you for clarification. I believe the results are important and deserved publication in a timely manner. With the amount of data presented, I mistakenly assumed this it was a full paper not a communication.
Author Response
Thank you for the positive evaluation.
Reviewer 3 Report
The current study focuses on the positioning of centrosome relative to the nucleus in migrating Dictyostelium discoideum cells by live-cell microscopy. All the set of results demonstrates that single cells preferred to position their centrosome at the back of the nucleus. I would like to thank the co-authors for answering some of my questions and comments, and for having improved many passages of the text. However, I am willing to admit that the deadline given by the editorial board to submit a revised version of their manuscript is very short to edit and collect supplementary results (even if the co-authors only have to ask to the editor for a deadline, which should grant them!). But it is unfortunate that many questions are not taken into account to further improve the quality of the presentation of their work. Thus, all of these preclude acceptance the article for publication in Cells, as presented below:
- reply to comment for the section Introduction: I agree with co-authors with the principle that “Dictyostelium is a model system that has been successfully used for decades to study general cell biological processes […]”, with some own biological aspects (notably concerning the centrosome). I do not question this point. However, by re-reading the introduction section, it remains very difficult to identify and understand the central question of the study that is not clearly written. Once again, I find results quite interesting and the quality of the work high.
- reply 1 “There are a number of excellent reviews, four of which are cited in our paper, that address this aspect. The point raised by this reviewer is not relevant to the central question of this paper. The present work is a Communication and not a Review”:
- comment: I agree with co-authors with the fact that the present work is a communication, and not a review. The question that the novice may ask is simply to understand why the co-authors succinctly address the involvement of G proteins in migratory processes, when this is not in the central question of the study. It is therefore useless to be so informative in the presentation if the authors do not wish to give more details.
- reply 2 “We do not ignore the importance of […]”: I validate the modifications / clarifications made in the text.
- reply 3 “The wording has been changed”:
- comment: thanks for clarifying the text.
- reply 4 “This manuscript has been submitted as part of a special issue dealing specifically with the function of centrosomes in various organisms. As part of this special issue, the group of Prof. Gräf published a paper summarizing the latest data and findings on centrosomes in Dictyostelium. This work is cited, and we deem it unnecessary to recapitulate these results in our communication”:
- comment: I understand that this manuscript is proposed for publication in a special issue dealing with the functions of centrosomes. However, I do not agree with the co-authors who would like to avoid recalling some crucial details on centrosomes on the pretext that another group will do so in the same issue. All articles in this special issue will appear independently of each other as search result in conventional web site (eg PubMed, etc.). Moreover, as mentioned in my general comment in the Introduction section above, it could be in this passage that the central question could be stated.
- replies 5-6: all clarifications in the section have been done. Thank you.
- reply 7 “Yes, it is the position of the nucleus in respect to the front and back of the cell as now shown in detail in the new Supplementary Figure S2”:
- comment: I validate the modifications / clarifications made in the text and figure S2
- reply 8 “Unfortunately, under the experimental conditions, not all cells develop into aggregation competent cells, and the rounded cells have not yet reached aggregation competence. We do not believe that the splitting into DIC and fluorescence images is helpful to better discern the position of centrosome and nucleus. We ask the reviewer to watch the corresponding movies as well”:
- comment: the coauthors replied to my comment, thank you.
- reply 9 “Thanks for your thoughts. The experiment suggested is a good idea but was not doable within the given time for revision”:
- comment: I understand that the authors do not wish to carry out supplemental experiments within the imposed deadline for revision. Here, my proposal to supplement thses data with nocodazole wash-out experiments, commonly conducted in this type of approach, is not crucial to support their conclusions. Maybe since the submission of the revised version of the manuscript, they collected these results, and I leave the freedom to the co-authors to add them or not to complete their study.
- reply 10 “No doubt that dynein and Kif9 are also of relevance in Dictyostelium. We do not understand what exactly this reviewer wants to tell us”:
- comment: my comment concerns the echoes of these results: it is well described that microtubules actively contribute to the correct orientation of the centrosome-nucleus axis in many cells models (see Januschke et al, Development, 2006; Zhao et al, Science, 2012; Carazo-Salas and Nurse, Nat Cell Biol, 2006; Vogl et al, Eur J Cell Biol, 1995, etc.). Does their originality reside in the Dictyostelium model?
- reply 11 “We think the classification of the centrosome in relation to the nucleus established in subcategories bring more details to our characterization of the centrosome position. Therefore, we would like to keep these sub-classifications of the side position into side-front and side-back”:
- comment: My comment was given to help the co-authors to simplify the representation of the data and to facilitate their understanding. The authors wish to maintain their choice of sub-classifications, which I validate.
- reply 12 “Dictyostelium cells feed on bacteria during their growth phase. Many probiotic bacteria produce folate which was detected to act as chemoattractant for Dictyostelium in the 1980s. In a Communication, we believe that it would be inappropriate to cover all the basic knowledge of decades of Dictyostelium research. We clearly state the fact that folate serves as chemoattractant”:
- comment: I do not propose to the authors to review all the knowledge on folate as chemoattractant, but a few reminders in one or two lines (supported by references) in the preamble to the results would help the novice in the topic. Please, rewrite this part.
- reply 13 “Thanks again for your thoughts. The experiments suggested are interesting, but not doable within the given time for revision. We may address these points in the future”:
- comment: obviously, this is more work than the given time for revision allows, I hope to read your obtained results from this point in the future.
- all the corrections reported with minor comments have been done. Thank you.
